# Application of Fourier Transform Infrared (FT-IR) Spectroscopy, Multispectral Imaging (MSI) and Electronic Nose (E-Nose) for the Rapid Evaluation of the Microbiological Quality of Gilthead Sea Bream Fillets

**DOI:** 10.3390/foods11152356

**Published:** 2022-08-06

**Authors:** Maria Govari, Paschalitsa Tryfinopoulou, Efstathios Z. Panagou, George-John E. Nychas

**Affiliations:** Laboratory of Microbiology and Biotechnology of Foods, Department of Food Science and Human Nutrition, School of Food and Nutritional Sciences, Agricultural University of Athens, Iera Odos 75, 11855 Athens, Greece

**Keywords:** gilthead sea bream fillets, FT-IR spectroscopy, electronic nose, multispectral imaging, modified atmosphere packaging, PLS-R

## Abstract

The potential of Fourier transform infrared (FT-IR) spectroscopy, multispectral imaging (MSI), and electronic nose (E-nose) was explored in order to determine the microbiological quality of gilthead sea bream (*Sparus aurata*) fillets. Fish fillets were maintained at four temperatures (0, 4, 8, and 12 °C) under aerobic conditions and modified atmosphere packaging (MAP) (33% CO_2_, 19% O_2_, 48% N_2_) for up to 330 and 773 h, respectively, for the determination of the population of total viable counts (TVC). In parallel, spectral data were acquired by means of FT-IR and MSI techniques, whereas the volatile profile of the samples was monitored using an E-nose. Thereafter, the collected data were correlated to microbiological counts to estimate the TVC during fish fillet storage. The obtained results demonstrated that the partial least squares regression (PLS-R) models developed on FT-IR data provided satisfactory performance in the estimation of TVC for both aerobic and MAP conditions, with coefficients of determination (R^2^) for calibration of 0.98 and 0.94, and root mean squared error of calibration (RMSE_C_) values of 0.43 and 0.87 log CFU/g, respectively. However, the performance of the PLS-R models developed on MSI data was less accurate with R^2^ values of 0.79 and 0.77, and RMSE_C_ values of 0.78 and 0.72 for aerobic and MAP storage, respectively. Finally, the least satisfactory performance was observed for the E-nose with the lowest R^2^ (0.34 and 0.17) and the highest RMSE_C_ (1.77 and 1.43 log CFU/g) values for aerobic and MAP conditions, respectively. The results of this work confirm the effectiveness of FT-IR spectroscopy for the rapid evaluation of the microbiological quality of gilthead sea bream fillets.

## 1. Introduction

The consumption of seafood products on a global basis amounts to 128 million tons, resulting in a per capita consumption of 18.4 kg/year of seafood products [1]. According to the report of the FAO on the State of World Fisheries and Aquaculture, fish products make up 15% of the intake of animal protein in 43 million people [2]. The increasing demand for fish consumption that has been seen in recent years, ranging from 130 to 150 million tons in the period 2011–2016, could be attributed to the high nutritive value of fish and also to the development of preservation techniques that can retain fish quality and allow marketing in different parts of the world [3].

Gilthead sea bream (*Sparus aurata*) is an important fish species farmed in the Mediterranean area. Greece is one of the largest producers of farmed gilthead sea bream, providing almost 26% of the annual world production [4]. Gilthead sea bream are commercialized in the European Union fish markets as whole and filleted, covered with ice, or maintained at refrigerated temperatures, mainly in aerobic conditions. Moreover, storage of fish under modified atmosphere packaging (MAP) is a common practice employed by the fish industry today to prolong the shelf life and preserve the quality characteristics of the product compared to aerobic storage [5,6].

Fish quality is mainly affected by indigenous microbiota and their metabolic activity (i.e., rapidly metabolized compounds), which results in loss of quality and freshness. Sheng and Wang [7] reported that pathogenic bacteria could contaminate fish at all stages of production, processing (e.g., stainless steel) [8], and the supply chain. Apart from bacteria, indigenous enzyme activity and oxidation of fish components, under specific conditions, could also contribute to fish spoilage [9]. Detection of microbial growth and control of microbial spoilage is an important issue in fish quality. Several methods have been broadly used for the evaluation of fish microbiological quality including culture methods and immunoassay-based or polymerase chain reaction (PCR) methods involving microbial DNA analysis [10]. It must be noted that these methods are laborious, time consuming, and cannot be implemented for the real-time detection of fish spoilage in a rapid and non-destructive manner [11]. Thus, the use of rapid, reliable, user-friendly, and non-destructive techniques for the determination of the microbiological quality and freshness of fish is of paramount importance for the fish industry, retailers, consumers, and inspection authorities.

In the last decade, various rapid analytical platforms e.g., FT-IR, near infra-red (NIR), MSI, and hyperspectral imaging (his), have been employed for the assessment of foods in terms of their microbiological quality [12,13]. The above-mentioned platforms in tandem with data analytics have been effectively employed in the estimation of the population of bacteria that cause spoilage and quality degradation in meat products [14,15,16,17,18] and more recently in fish [11,19]. FT-IR in conjunction with multivariate data analysis proved important for the quantification of spoilage bacteria in fish [19] and for the rapid prediction of fish fillet quality in terms of pH changes and chemical composition deterioration [20]. Multispectral imaging (MSI) is a rapid and non-destructive technology that requires no prior sample preparation, allowing the assessment of the microbiological quality in several foods, including fish, by combining spatial and spectral information [13,21]. An electronic nose (E-nose) is a biomimetic sensor equipped with an array of sensors with partial specificity combined with a pattern recognition system for the identification of food volatile compounds. The electronic nose is not focused on the detection of specific volatile compounds, but provides an ‘electronic volatile fingerprint’, which is characteristic of a specific food sample at a certain time [22]. The electronic nose has been also used successfully in the rapid evaluation of the microbiological quality of foods [17,23,24,25].

Therefore, the purpose of this work was to explore the effectiveness of Fourier transform infrared (FT-IR) spectroscopy, multispectral imaging (MSI), and an electronic nose (E-nose) in tandem with machine learning for the rapid evaluation of the microbiological quality of gilthead sea bream fillets stored aerobically and under MAP conditions at different temperatures.

## 2. Materials and Methods

### 2.1. Fish Fillet Storage and Sampling

Farmed gilthead sea bream (*Sparus aurata*) fillets (ca. 250 g each) were provided directly from Selonda Aquaculture S.A. The fillets were supplied in packs from two different fish batches and transferred to the laboratory in ice within 12 h of deboning. One batch was maintained under aerobic conditions (n = 112) and the second batch under MAP (33% CO_2_, 19% O_2_, 48% N_2_) (n = 112). In order to simulate refrigerated storage scenarios in the retail market, the gilthead sea bream fillets were stored at 0, 4, (refrigerated storage), 8, and 12 °C (temperature abuse) until spoilage was pronounced. Specifically, for aerobic conditions, the gilthead sea bream fillets were stored for 330 (n = 32) and 186 h (n = 30) at 0 and 4 °C, respectively, and for 126 h at 8 and 12 °C (n = 32 at each storage temperature). Under MAP conditions, the fish fillets were stored for 773 (n = 32), 473 (n = 30), 281 (n = 26), and 209 h (n = 30) at 0, 4, 8, and 12 °C, respectively. Duplicate samples of fish fillets were randomly taken from each storage temperature and packaging condition and subjected to microbiological analysis, sensory evaluation, FT-IR and MSI spectral data acquisition, and E-nose measurements. Analysis of fish fillets was performed upon arrival of the samples at the laboratory and at predetermined time slots according to storage temperature.

### 2.2. Microbiological Analysis

A portion of the dorsal ham of gilthead sea bream fillet (25 g) was homogenized with 225 mL of saline diluent (0.1%, w/v, peptone and 0.85%, w/v, NaCl) for 1 min at room temperature, using a stomacher device (Seward Medical, London, UK). Subsequently, serial decimal dilutions were prepared and 0.1 mL of appropriate dilutions were spread in duplicate on plate count agar (PCA, Biolife, Milano, Italy, 4021452) plates, for the enumeration of total viable counts after incubation at 25 °C for 72 h. The results were expressed as log CFU/g.

### 2.3. Sensory Assessment

During storage and at the same sampling points as for microbiological analyses and data acquisition, duplicate samples of fish fillets were assessed organoleptically by a five-member laboratory-trained sensory panel. Panelists were selected, trained, and checked according to ISO 8586-1 [26]. The sensory attributes assessed were the color of the skin and the odor of fish fillets using a five-point hedonic scale in the range of 1.0 (excellent quality, typical fresh odor, characteristic color) to 5.0 (non-acceptable quality, putrid odor, evident discoloration). Scores exceeding the value of 3.0 indicated the end of the gilthead sea bream fillets’ shelf life [27].

### 2.4. Spectral Data Acquisition

FT-IR spectral data were acquired from the skin of gilthead sea bream fillets using a ZnSe 45° HATR crystal (PIKE Technologies, Madison, WI, USA) and an FT-IR 6200 JASCO spectrometer (Jasco Corp., Tokyo, Japan), with a triglycine sulphate detector and a Ge/KBr beam splitter. The collected spectra were analyzed using the Spectra Manager™ Code of Federal Regulations (CFR) software version 2 (Jasco Corp.). The FT-IR spectral data over the wavenumber range of 3100–2700 cm^−1^ and 1800–900 cm^−1^ were selected for further analysis [11].

In addition, multispectral images from the skin of gilthead sea bream fillets were acquired using the Videometer Lab apparatus (Videometer A/S, Hørsholm, Denmark). This instrument acquires multispectral images in 18 non-uniformly distributed wavelengths ranging from 405 to 970 nm [11,28]. The advantage of this method is that it provides information in the visible and short NIR region and, at the same time, it uses the spatial information of each pixel. Image acquisition, segmentation, and model development have been detailed previously [12,29]. After the analysis of the images, feature extraction included the mean reflectance values of the 18 wavelengths (±the standard deviation) that were further assessed using multivariate analysis.

### 2.5. E-Nose Measurements

The volatile profile of the fish samples was also monitored using a FOX 3000 electronic nose (Alpha M.O.S., Toulouse, France) equipped with 12 metal oxide sensors (Table 1), an injection system, a mass flow controller, and pattern recognition software (Alpha Soft V14). A portion of fish sample (ca. 2 g) was transferred into a 20 mL volume glass vial, sealed with a PTFE/silicone septum and aluminum screw cap, and heated at 50 °C for 20 min in a thermoblock static headspace sampler to generate the headspace volatiles. A volume of 0.5 mL of the headspace was injected into the E-nose and the volatiles were measured as sensor resistance changes over time:(1)$$\Delta R=\frac{R_{t}-R_{0}}{R_{0}}$$
where R_t_ is the resistance of the sensor at time *t* and R_0_ is the baseline resistance (*t* = 0). The acquisition time was set to 120 s, which was followed by a recovery period of 1080 s so that the sensors returned to the baseline. The maximum sensor resistance was employed for data analysis. Details of the operating conditions of the E-nose can be found elsewhere [30].

### 2.6. Data Analysis

Partial least squares regression (PLS-R) models were developed and validated for the estimation of the microbial load of gilthead sea bream fillets. The underlying principle of the analysis was to explore the feasibility to predict TVC directly from the acquired spectral and volatile fingerprints during storage of fish fillet samples regardless of storage temperature. For this reason, FT-IR and MSI spectral data and E-nose data were employed as exploratory (independent) variables and TVC as the target (dependent) variable. Specifically, in the case of MAP, model calibration was performed with FT-IR data obtained from fish samples at 0, 4, and 8 °C (n = 84), and model prediction was implemented using the data from the samples stored at 12 °C (n = 28). No preprocessing was applied to the data prior to analysis. In the case of aerobic storage, model calibration was based on FT-IR data derived from fish fillet samples stored at 0, 4, and 12 °C (n = 84), whereas model prediction was performed using the data obtained from samples stored at 8 °C (n = 28). Before analysis, data were transformed using the standard normal variate (SNV) [31].

In addition, in both aerobic and MAP packaging, MSI spectral data from fish fillet samples maintained at 0 and 4 °C (n = 59) were employed in PLS-R model calibration, whereas model prediction was undertaken with data obtained from samples at 8 and 12 °C (n = 54). No preprocessing was applied to MSI spectral data prior to analysis.

Finally, regarding the E-nose, for both aerobically and MAP packaged fish fillet samples, model calibration was performed with data obtained at refrigerated temperatures (0 and 4 °C) (n = 46), while prediction was performed with data derived at 8 and 12 °C (n=48). No preprocessing was applied to the E-nose data prior to analysis. Slope, offset, root mean squared error of calibration (RMSEc), root mean squared error of cross-validation (RMCEcv), root mean squared error of prediction (RMSEp), and the coefficients of determination (R^2^) for calibration, cross-validation, and prediction were the main indices employed in the evaluation of the applicability of the models. Generally, good models present high values of R^2^ and low values of RMSE. The optimum number of latent variables (LVs) was assigned at the minimum prediction residual error sum of squares (PRESS) after leave-one-out cross-validation (LOOCV) during model calibration to avoid overfitting [32]. The Unscrambler software ver. 9.7 (CAMO Software AS, Oslo, Norway) was used for data analysis.

## 3. Results and Discussion

### 3.1. Fish Quality Degradation Due to Microbial Growth

The TVC changes of gilthead sea bream fillets in both (aerobic and MAP) storage conditions are illustrated in Figure 1. The initial population of TVC was ca. 4.9 and 4.2 log CFU/g in aerobically and MAP packaged fish fillet samples, respectively. The growth profile of TVC was affected by storage temperature, resulting in higher populations with increasing storage temperature. In addition, higher populations were attained in aerobic storage compared to MAP. In the end of storage, the population of TVC ranged between 9.4–9.9 and 8.8–9.8 log CFU/g for aerobically and MAP packaged fish fillet samples, respectively. These results are in line with other researchers [33] who reported a similar growth profile of TVC for gilthead sea bream fillets stored at refrigerated temperatures (0, 5, and 15 °C) under both aerobic or MAP conditions (CO_2_ 60%, O_2_ 10% and N_2_ 30%). Indeed, the final populations of TVC were found in the range 8.3–8.9 and 7.5–8.1 log CFU/g for gilthead sea bream fillets stored under aerobic and MAP conditions, respectively.

### 3.2. Sensory Evaluation

The sensory evaluation of gilthead sea bream fillets kept under aerobic and MAP conditions is demonstrated in Figure 2 and Figure 3, respectively. The odor and color attributes showed higher scores with increasing storage temperature (loss of organoleptic characteristics) in both conditions. Fish fillets retained higher scores of organoleptic attributes for a longer time under MAP compared to aerobic conditions. According to the sensory evaluation, the sea bream fillets reached the rejection limit (score 3) under aerobic storage on 144, 78, 54, and 42 h at 0, 4, 8, and 12 °C, respectively, whereas for MAP storage, the same rejection limit was reached on 401, 257, 113, and 54 h at 0, 4, 8, and 12 °C, respectively. Sensory rejection coincided with TVC counts of ca. 7.2 and 7.4 log CFU/g for aerobic and MAP conditions, respectively. The organoleptic rejection of fish could be attributed to the metabolic compounds produced by the dominant microorganisms at these population levels [1,19,33].

### 3.3. Rapid Assessment of Fish Spoilage Using FT-IR, MSI, and E-Nose

Typical FT-IR spectra of the skin of gilthead sea bream fillets for fresh (TVC 4.95 log CFU/g) and spoiled (TVC 8.43 log CFU/g) samples stored aerobically and under MAP are illustrated in Figure 4. The TVC value for spoiled samples coincided with a storage period of 54 h at 8 °C and 112 h at 12 °C in air and MAP conditions, respectively. The FT-IR spectra in the approximate wavenumber ranges of 3100–2700 cm^−1^ and 1800–900 cm^−1^ provided information about the biochemical compounds resulting from microbial metabolism [11]. The peak at 1640 cm^−1^ (O-H stretch) is associated to water and amide I. The peaks at 1545 cm^−1^ (N-H bend, C-N stretch) and 1314–1238 cm^−1^ (C-N stretch, C=O-N bend and N-H bend) are ascribed to amide II and amide III, respectively. The peaks at 1162–1025 cm^−1^ (C-N stretch) are also associated to amines. It is also important to note that most of the above-mentioned peaks could be associated with the proteolytic activity of microorganisms during fish storage [34]. In addition, representative spectra from MSI and E-nose signals are provided in Appendix A, respectively. 

The results of the PLS-R models developed on FT-IR data for the estimation of the microbial population on all samples are depicted graphically by the comparison of the observed versus estimated TVC values in Figure 5 and Figure 6, respectively. The data points were uniformly located above and below the line of equity (y = x) and they were included in the ± 1.0 log unit area, which is acceptable from the microbiological perspective. In addition, the performance of the developed models for calibration, cross-validation, and prediction is summarized in Table 2. Results indicated a good correlation between FT-IR spectra and TVC. Specifically, for the gilthead sea bream fillets stored in air, the values of R^2^ were 0.98, 0.89, and 0.74 for model calibration, cross-validation, and prediction, respectively. Moreover, for the gilthead sea bream fillets stored under MAP, the respective values of R^2^ were 0.94, 0.76, and 0.94. In addition, the RMSEp values were low, namely, 0.87 log CFU/g and 0.43 log CFU/g for the sea bream fillets stored aerobically and under MAP, respectively. A high R^2^ value in association with low values of RMSE of calibration, cross-validation, and prediction indicate good performance of PLS-R models [35]. Thus, the performance metrics of the models lead to the conclusion that they are suitable to be applied to the direct prediction of the quality of bream fillets directly from FT-IR spectra regardless of storage temperature.

The concept of using FT-IR for the immediate determination of the quality level in fish in combination with machine learning is quite recent. Govari et al. [19] investigated the microbiological quality of farmed sea bass (*Dicentrarchus labrax*) fillets maintained aerobically and under MAP conditions at 0, 4, 8, and 12 °C using FT-IR spectroscopy combined with data analytics, taking into account the measured TVC populations. The developed PLS-R models performed well in the prediction of TVC with R^2^ values of 0.78 and 0.99 for aerobic and MAP conditions, respectively. In addition, Fengou et al. [11] investigated the application of FT-IR spectroscopy and multivariate data analysis for the estimation of the quality of farmed whole ungutted gilthead sea bream (*Sparus aurata*). The authors reported that PLS-R models created by measurements acquired from the fish skin resulted in a satisfactory prediction of TVC with R^2^ and RMSE_p_ values of 0.727 and 0.717 log CFU/g, respectively.

The results of the PLS-R models developed on MSI data are presented in Figure 5 and Figure 6, while the performance metrics of the models are included in Table 3. It is evident that the performance of the PLS-R models based on MSI data was less satisfactory compared to FT-IR. Specifically, the fillets stored aerobically showed R^2^ values of 0.79, 0.52, and 0.58 for model calibration, cross-validation, and prediction, respectively. Moreover, the respective R^2^ values for the samples packaged under MAP were 0.77, 0.60, and 0.54. Furthermore, the calculated RMSE_p_ values for TVC were 1.10 log CFU/g and 1.43 log CFU/g for fish fillet samples stored under MAP and air, respectively. In agreement with the present work, low R^2^ values were reported by other researchers for PLS-R models developed on MSI measurements obtained from the skin of sea bass fillets, stored under the same temperatures and packaging conditions [19]. Similarly, less satisfactory predictions were obtained with PLS-R models based on MSI data collected from the skin of whole ungutted sea bream fish under aerobic storage at 0, 4, and 8 °C, with R^2^ values of 0.589, 0.460, and 0.315, and RMSE values of 0.927, 1.074, and 1.136 (log CFU/g) for model development, cross-validation, and prediction, respectively [11]. A possible explanation for the low model performance could be attributed to the fact that the multispectral image was acquired on the skin of the fish that presented high reflectance and could thus affect the quality of the obtained information. However, according to previous studies, multispectral imaging combined with PLS-R model development presented satisfactory performance in the assessment of the microbiological quality of other foods, such as pork meat stored under air or MAP [11,28], poultry products, and beef fillets during aerobic storage [29].

The outcome of the PLS-R models developed on E-nose sensor array measurements for the assessment of TVC of gilthead sea bream fillets is illustrated in Figure 7, while the performance metrics of the models are shown in Table 4. It can be inferred that the performance of the PLS-R models developed on E-nose measurements was less satisfactory in the evaluation of the microbiological quality of fish fillets. Specifically, fish samples stored under aerobic conditions presented very low values of R^2^, namely, 0.17, 0.14, and 0.34 for model calibration, cross-validation, and prediction, respectively. The same was observed under MAP storage, where the respective R^2^ values were 0.21, 0.14, and 0.17. Furthermore, the estimated RMSEp values were 1.77 log CFU/g and 1.43 log CFU/g for the fish samples stored under MAP and air, respectively, again indicating less satisfactory performance. The development of an E-nose instrument equipped with sensors presenting high sensitivity to specific volatile compounds of fish such as trimethylamine and/or certain aldehydes and ketones could be more effectively used in the prediction of fish quality. Semeano et al. [36] developed a gas sensor equipment and successfully monitored the mesophilic bacterial counts of Tilapia fish by checking the headspace gases during storage at 20 °C. In addition, fish species differentiation was successfully performed using an electronic nose [37], whereas, recently, the freshness of red mullet, sole, and cuttlefish was successfully determined using a low-cost E-nose comprised of four metal oxide semiconductor (MOS) sensors [38]. In general, the implementation of an E-nose to determine the freshness of fish has attracted the attention of researchers [24,39,40] and in many cases MOS sensors have been used in E-nose instruments as they are readily available in the market and provide rapid response and good sensitivity to the presence of volatiles [41].

## 4. Conclusions

The findings of this work demonstrate that PLS-R models developed on FT-IR data for the evaluation of the microbiological quality of gilthead sea bream fillets packaged under aerobic and MAP conditions presented satisfactory performance in the prediction of microbial growth in terms of the TVC. In contrast, PLS-R models developed on MSI and E-nose data were less effective in the estimation of the TVC of fish samples stored in both conditions. Our results underline the effectiveness of FT-IR as a rapid and non-invasive technique for the assessment of microbial growth of gilthead sea bream fillets during refrigerated storage under MAP and aerobic conditions, which could thus become a valuable tool for the fish industry to evaluate product quality.

## Figures and Tables

**Figure 1 foods-11-02356-f001:**
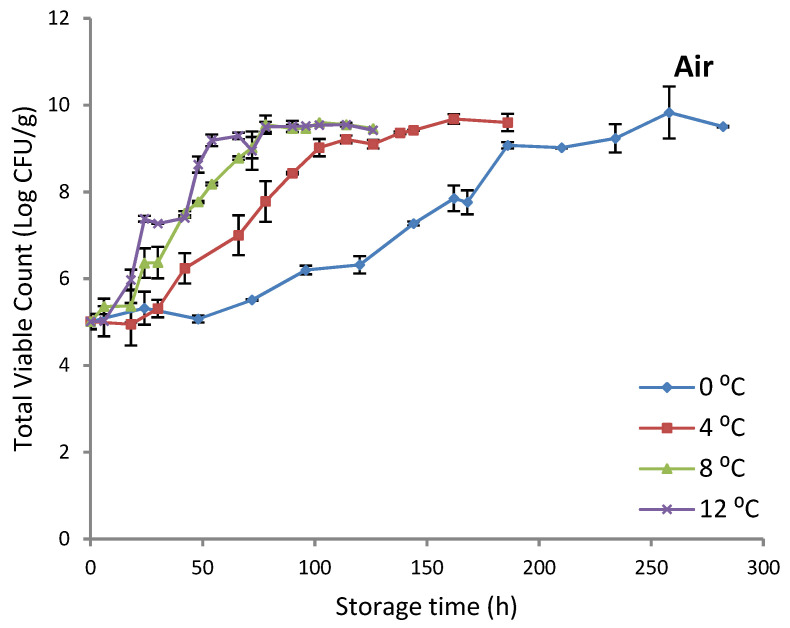
Changes in the population of total viable counts (TVC) during aerobic and MAP storage of gilthead sea bream fillets. Data points represent mean values ± standard deviation from duplicate packages analyzed per sampling point.

**Figure 2 foods-11-02356-f002:**
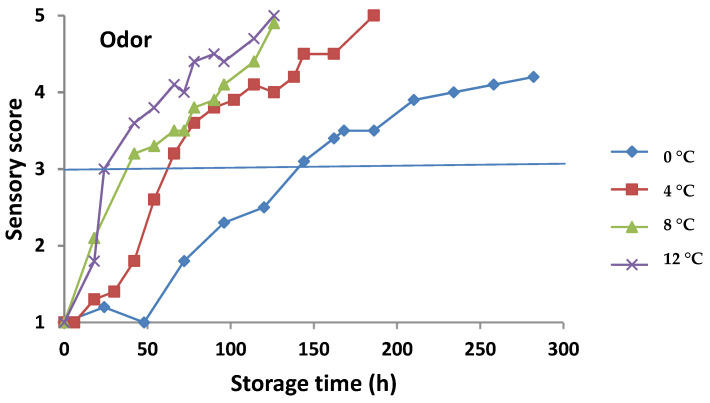
Sensory scores of gilthead sea bream fillets for odor and skin color attributes stored under aerobic conditions. Solid line indicates the threshold value for sample rejection.

**Figure 3 foods-11-02356-f003:**
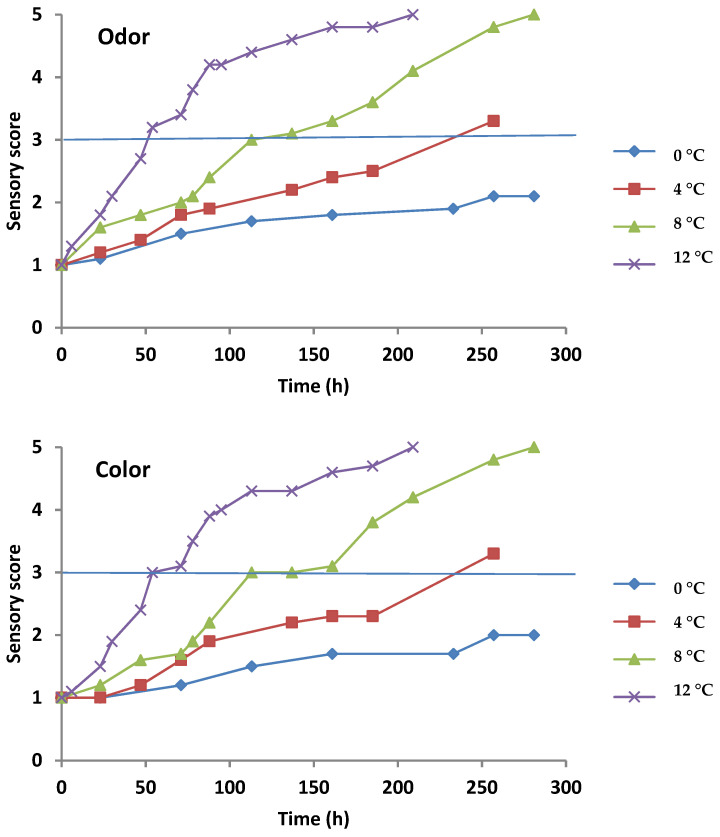
Sensory scores of gilthead sea bream fillets for odor and skin color attributes stored under MAP conditions. Solid line indicates the threshold value for sample rejection.

**Figure 4 foods-11-02356-f004:**
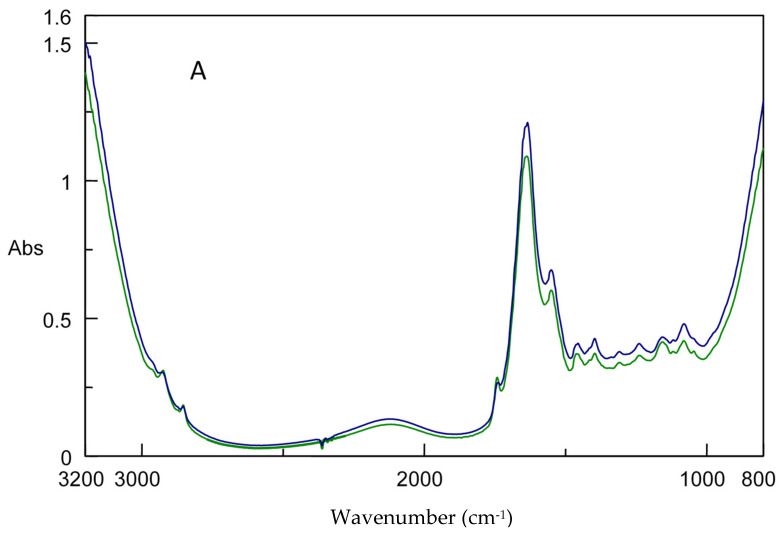
Representative FT-IR spectra corresponding to fresh (blue line, 4.95 log CFU/g) and spoiled (green line, 8.43 log CFU/g) gilthead sea bream fillet samples under aerobic (**A**) and MAP (**B**) storage conditions.

**Figure 5 foods-11-02356-f005:**
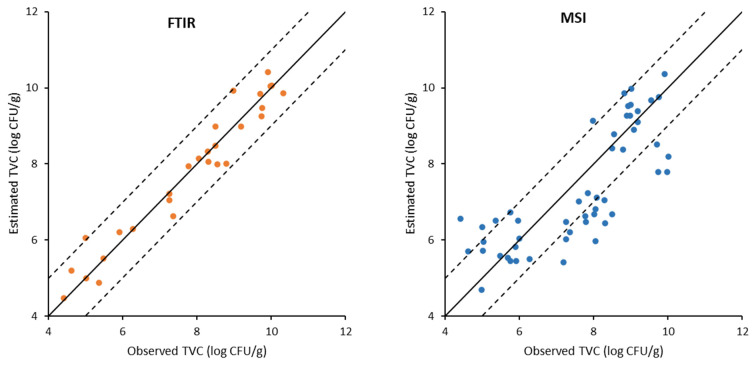
Correlation of observed and estimated values of total viable counts (TVC) on gilthead sea bream fillets during storage under MAP conditions, generated by the PLS-R model based on FT-IR and MSI data. Solid line indicates the line y = x (equity); dashed lines indicate deviation of ± 1 log unit.

**Figure 6 foods-11-02356-f006:**
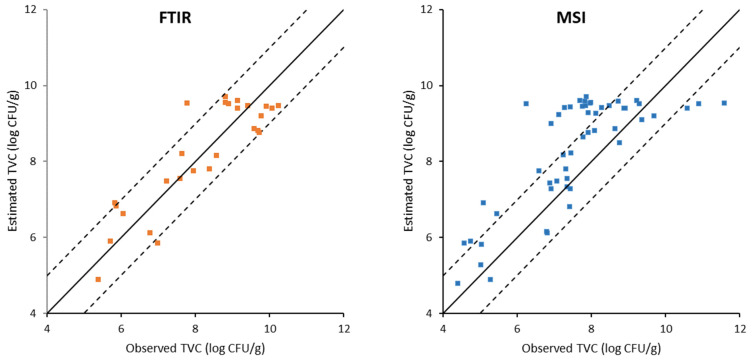
Correlation of observed and estimated values of total viable counts (TVC) on gilthead sea bream fillets during aerobic storage, generated by the PLS-R models based on FT-IR and MSI data. Solid line indicates the line y = x (equity); dashed lines indicate deviation of ± 1 log unit.

**Figure 7 foods-11-02356-f007:**
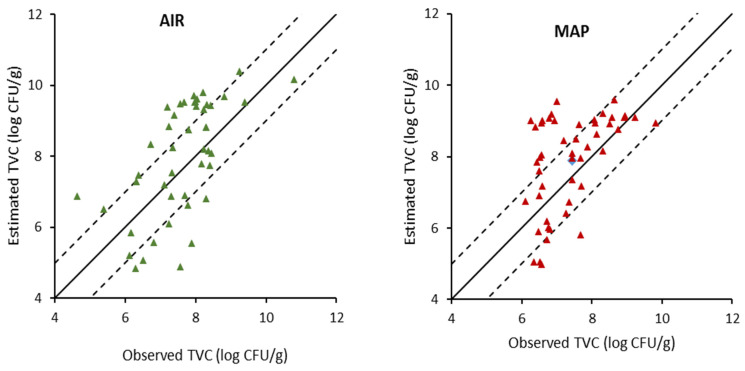
Correlation of observed and estimated values of total viable counts (TVC) on gilthead sea bream fillets during storage in air and MAP conditions, generated by the PLS-R models based on E-nose data. Solid line indicates the line y = x (equity); dashed lines indicate deviation of ± 1 log unit.

**Table 1 foods-11-02356-t001:** Types of FOX 3000 electronic nose sensors and their specificity for chemical compounds.

Sensor Number	Name	Detection of Chemical Components
1	LY/LG	Oxidation gas
2	LY2/G	NH_3_/CO
3	LY2/AA	C_2_H_5_OH
4	LY2/GH	NH_3_/Amine
5	LY2/gCTL	H_2_S
6	LY2/gCT	C_3_H_8_/C_4_H_10_
7	T30/1	Organic solvents
8	P10/1	Hydrocarbons
9	P10/2	CH_4_
10	P10/2	F_2_
11	T70/2	Aromatic components
12	PA/2	C_2_H_5_OH/NH_3_/Amine

**Table 2 foods-11-02356-t002:** Performance metrics of the PLS-R models developed on FT-IR data for the estimation of TVC in gilthead sea bream fillet samples.

Storage	Data Set	LV	Slope	Offset	R^2^	RMSE
Air	Calibration	7	0.98	0.10	0.98	0.16
	Cross-validation *		0.94	0.50	0.89	0.49
	Prediction		0.78	1.80	0.74	0.87
MAP	Calibration	7	0.94	0.38	0.94	0.38
	Cross-validation *		0.80	1.35	0.76	0.78
	Prediction		0.94	0.44	0.94	0.43

R^2^: coefficient of determination; RMSE: root mean squared error; * Leave-one-out cross-validation, X_explained variance_: 91%, Y_explained variance_: 52% (aerobic conditions); X_explained variance_: 78%, Y_explained variance_: 20% (MAP).

**Table 3 foods-11-02356-t003:** Performance metrics of the PLS-R models developed on MSI data for the estimation of TVC in gilthead sea bream fillet samples.

Storage	Data Set	LV	Slope	Offset	R^2^	RMSE
Air	Calibration	9	0.79	1.59	0.79	0.78
	Cross-validation *		0.67	2.54	0.52	1.21
	Prediction		0.88	0.36	0.58	1.43
MAP	Calibration	9	0.77	1.42	0.77	0.72
	Cross-validation *		0.72	1.77	0.60	0.97
	Prediction		0.80	1.24	0.54	1.10

R^2^: coefficient of determination; RMSE: root mean squared error; * Leave-one-out cross-validation, X_explained variance_: 97%, Y_explained variance_: 31% (aerobic conditions); X_explained variance_: 78%, Y_explained variance_: 44% (MAP).

**Table 4 foods-11-02356-t004:** Performance metrics of the PLS-R models developed on E-nose measurements for the estimation of TVC in gilthead sea bream fillet samples.

Storage	Data Set	LV	Slope	Offset	R^2^	RMSE
Air	Calibration	3	0.21	6.04	0.21	1.47
	Cross-validation *		0.18	6.28	0.14	1.56
	Prediction		0.16	6.38	0.17	1.43
MAP	Calibration	3	0.17	5.50	0.17	1.54
	Cross-validation *		0.16	5.61	0.14	1.59
	Prediction		0.34	6.31	0.34	1.77

R^2^: coefficient of determination; RMSE: root mean squared error; * Leave-one-out cross-validation, X_explained variance_: 76%, Y_explained variance_: 17% (aerobic conditions); X_explained variance_: 95%, Y_explained variance_: 7% (MAP).

## Data Availability

The data presented in this study are available on request from the corresponding author. The data are not publicly available due to privacy.

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
