# Peer review of "Application of Fourier Transform Infrared (FT-IR) Spectroscopy, Multispectral Imaging (MSI) and Electronic Nose (E-Nose) for the Rapid Evaluation of the Microbiological Quality of Gilthead Sea Bream Fillets"

_foods, 2022, doi:10.3390/foods11152356_

Round 1
Reviewer 1 Report
The paper uses FT-IR spectroscopy, MSI and E-nose to evaluate e the microbiological quality of gilthead sea bream (Sparus aurata) fillets. This work belongs to the application of nondestructive testing technology for food quality, and it has certain practical significance. The points generating more doubts are:
1. Abstract and body parts (separate), the first occurrence of an acronym should be given its full name. For example: Abstract (Line 20: PLS-R, Line 24: RMSEc, etc.). The same goes for the body part.
2. The review of FT-IR, MSI and E-nose in fish microbiological evaluation is not comprehensive enough (introduction should be supplemented).
3. The description of the sample preparation section is very unclear. How many samples were taken each time under different storage conditions (aerobic conditions, MAP conditions), different temperatures and storage times? How many samples were obtained in total for instrumental data collection, sensory analysis, and physicochemical analysis? These contents should be explained clearly in Section 2.1.
4. In the sensory evaluation of the samples, the authors should include detailed information about the sensory methodology, including how assessors were selected and trained, attribute definition, the references used for attribute identification and scoring, and how panel performance was checked.
5. In terms of instrument data presentation, MSI and E-nose data were not seen in the paper. If it is not convenient for the author to give it in the main text, it is recommended that the author describe this part of the data in the form of an attachment, so that the reader can understand the acquisition of instrument data of the sample more clearly.
6. During data analysis, the author did not divide the sample set reasonably, because the samples in the calibration set could not contain the characteristics of the samples in the prediction set. This is unreasonable from a chemometric analysis point of view. In addition, in the analysis of FT-IR spectral data, the spectra obtained under MAP storage conditions were not preprocessed, while those obtained under aerobic storage conditions were preprocessed. Why? And the sample set division in these two modes is also inconsistent, why? Is the conclusion obtained by comparing the results of the PLS model obtained in this way objective? I am very confused about this part.
Similarly, for the analysis of MSI data, only spectral data is used, or are both spectral and image data applied? Are spectra preprocessed? When analyzing the E-nose data, do you use part of the characteristic data or all the E-nose signal data? None of these are detailed in the paper.
7. The PLS study is highly incomplete. A usual PL model report should include: (a) optimum number of latent variables, (b) methods for estimating the latter number (leave-one-out cross-validation, Monte Carlo cross-validation, randomization tests, independent monitoring test, etc.), (c) explained variance in X and Y blocks, etc.
8. The title suggests appropriate revisions by the author. Because the author uses the E-nose technology to obtain sample data, the headspace gas enrichment time takes 20 minutes, which can no longer be regarded as a rapid evaluation.
Author Response
Reviewer 1
The paper uses FT-IR spectroscopy, MSI and E-nose to evaluate e the microbiological quality of gilthead sea bream (Sparus aurata) fillets. This work belongs to the application of nondestructive testing technology for food quality, and it has certain practical significance. The points generating more doubts are:
Comment 1: Abstract and body parts (separate), the first occurrence of an acronym should be given its full name. For example: Abstract (Line 20: PLS-R, Line 24: RMSEc, etc.). The same goes for the body part.
Response: Your comment was adopted and the text was modified accordingly by providing the full name of the acronyms in the abstract and throughout the main text.
Comment 2: The review of FT-IR, MSI and E-nose in fish microbiological evaluation is not comprehensive enough (introduction should be supplemented).
Response: Your comment was adopted and the text has been modified accordingly.
Comment 3: The description of the sample preparation section is very unclear. How many samples were taken each time under different storage conditions (aerobic conditions, MAP conditions), different temperatures and storage times? How many samples were obtained in total for instrumental data collection, sensory analysis, and physicochemical analysis? These contents should be explained clearly in Section 2.1.
Response: The requested information has been included in the revised section 2.1.
Comment 4: In the sensory evaluation of the samples, the authors should include detailed information about the sensory methodology, including how assessors were selected and trained, attribute definition, the references used for attribute identification and scoring, and how panel performance was checked.
Response: Your comment was adopted and the text was modified accordingly. See revised paragraph 2.3.
Comment 5: In terms of instrument data presentation, MSI and E-nose data were not seen in the paper. If it is not convenient for the author to give it in the main text, it is recommended that the author describe this part of the data in the form of an attachment, so that the reader can understand the acquisition of instrument data of the sample more clearly.
Response: The requested data from MSI and E-nose instruments are provided as Supplementary Figures.
Comment 6: During data analysis, the author did not divide the sample set reasonably, because the samples in the calibration set could not contain the characteristics of the samples in the prediction set. This is unreasonable from a chemometric analysis point of view. In addition, in the analysis of FT-IR spectral data, the spectra obtained under MAP storage conditions were not preprocessed, while those obtained under aerobic storage conditions were preprocessed. Why? And the sample set division in these two modes is also inconsistent, why? Is the conclusion obtained by comparing the results of the PLS model obtained in this way objective? I am very confused about this part.
Similarly, for the analysis of MSI data, only spectral data is used, or are both spectral and image data applied? Are spectra preprocessed? When analyzing the E-nose data, do you use part of the characteristic data or all the E-nose signal data? None of these are detailed in the paper.
Response: It is indeed true that from a chemometric analysis perspective it is better to join the data from all storage temperatures and make a spit of e.g. 80/20 for training and testing. This process has been implemented by a previous work undertaken in our lab (Fengou et al., 2019 as cited in the reference list). However, in practice the temperature pre-history of a food sample is not known and thus it is necessary to develop the model and validate the model under other storage temperatures. In fact, we trained the model with data obtained from 0 and 4°C (which are the normal storage temperatures of fish) and validated the model under temperature abuse conditions (8 and 12°C). The reason is that we wanted to test the performance of the model under extreme conditions for fish storage. Concerning the different preprocessing methods, it must be stated that the data were preprocessed with the same methods and only the best results are presented in the final manuscript. In this sense, FT-IR spectral data presented better results in PLS-R model development without preprocessing in MAP packaged fish fillets, compared with aerobically packaged fish fillets, where the PLS-R model developed on FT-IR data presented the best performance after SNV transformation.
Concerning MSI data, the feature extraction after the analysis of images contains the reflectance values of the 18 wavelengths from the Videometer instrument. This information has been added in the revised paragraph 2.4. No pre-processing of the data was undertaken prior to analysis. As far as E-nose analysis is concerned, the maximum of sensor resistance was employed in data analysis. This information is included in the revised manuscript.
Comment 7: The PLS study is highly incomplete. A usual PL model report should include: (a) optimum number of latent variables, (b) methods for estimating the latter number (leave-one-out cross-validation, Monte Carlo cross-validation, randomization tests, independent monitoring test, etc.), (c) explained variance in X and Y blocks, etc.
Response: The number for estimating the latent variables is indicated in the last paragraph of data analysis section, which is leave-one-out cross validation. The remaining information about the number of latent variables and the explained variance in X and Y blocks has been included in the revised Tables 2, 3 and 4.
Comment 8: The title suggests appropriate revisions by the author. Because the author uses the E-nose technology to obtain sample data, the headspace gas enrichment time takes 20 minutes, which can no longer be regarded as a rapid evaluation.
Response: It is indeed true that the development of the headspace prior to the analysis with the E-nose takes about 20 minutes, but the analysis itself with the E-nose takes 2 min. In this sense it is a rapid analysis. In fact, we compare these instrumental techniques with conventional microbiological analysis using the standard plating technique that takes about 2-3 days to get the results for the microorganisms. In this sense all other instrumental analysis could be considered rapid even if we need 20 min for the development of headspace. So, the comparison must be made with standard microbiological measurements. So, if the reviewer has no objection and agrees with our perspective, we would like to maintain the word “rapid” in the title of this manuscript.
Reviewer 2 Report
This work has explored the effectiveness of Fourier transform infrared (FT-IR) spectroscopy, multispectral imaging (MSI) and electronic nose (E-nose) in tandem with machine learning, for the evaluation of the microbiological quality of gilthead sea bream fillets stored aerobically and under MAP conditions at different temperatures. However, and the paper as a whole lacks more convincing proof of the stability and sensitivity of this method. So, some issues should be addressed and I think this article needs a major revision work.
1. Line 153-160: In the case of MAP, no preprocessing was applied to the data prior to analysis. While in the case of aerobic storage, the analysis data were transformed using the standard normal variate (SNV) before. Why is it designed this way, please explain?
2. Line 153-160: Under aerobic storage and MAP conditions, why are the temperatures of 0oC, 4oC, 8oC, 12oC respectively selected? How long are these samples stored separately? Please explain?
3. Line 167-170: It is necessary to further clarify the significance of different indicators and their influence on the stability of the model.
4. Before using the PLS-R model, have you considered using the corresponding variable screening method to increase the stability of the model?
5. Please carefully check and modify the format of the chart in the article to ensure that it presents the best image and meets all technical requirements of the Journal of foods.
6. PLS-R models developed on FT-IR data for the evaluation of the microbiological quality of gilthead sea bream fillets presented satisfactory performance in the prediction of the microbial growth in terms of TVC, but the paper does not mention the low detection limit of this method, and how does its low detection limit compare with standard methods?
7. There has been a great deal of literature reported that using FTIR spectroscopy coupled with machine learning for the rapid evaluation of the microbiological quality of fish, I think you should focus on explaining how your work is better or more innovative than previous research.
Author Response
Reviewer 2
This work has explored the effectiveness of Fourier transform infrared (FT-IR) spectroscopy, multispectral imaging (MSI) and electronic nose (E-nose) in tandem with machine learning, for the evaluation of the microbiological quality of gilthead sea bream fillets stored aerobically and under MAP conditions at different temperatures. However, and the paper as a whole lacks more convincing proof of the stability and sensitivity of this method. So, some issues should be addressed and I think this article needs a major revision work.
Comment 1: Line 153-160: In the case of MAP, no preprocessing was applied to the data prior to analysis. While in the case of aerobic storage, the analysis data were transformed using the standard normal variate (SNV) before. Why is it designed this way, please explain?
Response: This issue has been raised by the Reviewer 1 and it has been tacked before. Please see our response to the first reviewer.
Comment 2: Line 153-160: Under aerobic storage and MAP conditions, why are the temperatures of 0oC, 4oC, 8oC, 12oC respectively selected? How long are these samples stored separately? Please explain?
Response: The former two temperatures correspond to refrigerated storage of fish, whereas the latter two temperatures correspond to temperature abuse that may happen in the distribution chain. A clarification on this issue has been added in the revised paragraph 2.1.
Comment 3: Line 167-170: It is necessary to further clarify the significance of different indicators and their influence on the stability of the model.
Response: Generally, the higher the values of R2 and the lower the values of RMSE, the better the model.
Comment 4: Before using the PLS-R model, have you considered using the corresponding variable screening method to increase the stability of the model?
Response: No, we have not taken into consideration such approach.
Comment 5: Please carefully check and modify the format of the chart in the article to ensure that it presents the best image and meets all technical requirements of the Journal of foods.
Response: All possible efforts have been undertaken to comply with the requirements of the Journal.
Comment 6: PLS-R models developed on FT-IR data for the evaluation of the microbiological quality of gilthead sea bream fillets presented satisfactory performance in the prediction of the microbial growth in terms of TVC, but the paper does not mention the low detection limit of this method, and how does its low detection limit compare with standard methods?
Response: This is indeed a good comment, but this aspect was beyond the scope of our research.
Comment 7: There has been a great deal of literature reported that using FTIR spectroscopy coupled with machine learning for the rapid evaluation of the microbiological quality of fish, I think you should focus on explaining how your work is better or more innovative than previous research.
Response: The innovative part in this work lies in the fact that we did not use only FT-IR but we compared the effectiveness of three rapid techniques (FT-IR, MSI and E-nose) in the evaluation of the microbiological quality of fish fillets. To the best of our knowledge, this approach is employed for the first time in fish fillets.
Reviewer 3 Report
The experiment and analyze were appropriately conducted. However, there are several minor and major problems that need to be addressed.
1. (Page 1, Line 22) The obtained results demonstrated that the PLS-R models developed on FT-IR data, with coefficients of determination (R 2 ) of 0.98 and 0.94 , is not in conformity with the Table 2.
2. (Page 4, Line 19) The spectral data acquisition does not provide the number of golden head snappers are in the correction set, validation set and prediction set, and the data set division method is not mentioned.
3. (Page 4, Table 2) If the authors provide preprocessing that is applied to the data prior to analysis or use other regression models, whether you can get better results, because PLS-R does not be suitable for nonlinear data.
4.(Page 9, Line 8) The paper mentions the Figures 5A and 6A, the authors don’ t indicate in the Figures 5 and 6.
5. (Page 10, Line 12) The results and discussion does not provide the original spectra of MSI and only adds the table of values.
6. (Page 10, Line 29) The respective R2 values for the samples packaged under MAP were 0.77, 0.60, and 0.56 is not in conformity with the Table 3.
7. (Page 12, Table 2) The authors referred to the findings of this work demonstrated that PLS-R models developed on FT-IR data for the evaluation of the microbiological quality of gilthead sea bream fillets packaged under aerobic and MAP conditions presented satisfactory performance in the prediction of the microbial growth in terms of TVC, but the respective R2 value is much lower than the cross-validation R2 value.
Author Response
Reviewer 3
The experiment and analyze were appropriately conducted. However, there are several minor and major problems that need to be addressed.
Comment 1: (Page 1, Line 22) The obtained results demonstrated that the PLS-R models developed on FT-IR data, with coefficients of determination (R 2 ) of 0.98 and 0.94 , is not in conformity with the Table 2.
Response: These values of R2 correspond to the calibration procedure of the PLS-R model. A clarification has been added in the abstract to avoid confusion.
Comment 2: (Page 4, Line 19) The spectral data acquisition does not provide the number of golden head snappers are in the correction set, validation set and prediction set, and the data set division method is not mentioned.
Response: All these information are presented in the revised paragraph 2.1 and 2.6 of the Materials and Methods section.
Comment 3: (Page 4, Table 2) If the authors provide preprocessing that is applied to the data prior to analysis or use other regression models, whether you can get better results, because PLS-R does not be suitable for nonlinear data.
Response: We have used only PLS-R model development as a golden standard in this kind of analysis. The non-linear nature of the data cannot be excluded and thus the use of other regression models could provide better performance. However, this was beyond the scope of this article. In addition, the preprocessing methods applied to the dataset are included in the revised paragraph 2.6.
Comment 4: (Page 9, Line 8) The paper mentions the Figures 5A and 6A, the authors don’t indicate in the Figures 5 and 6.
Response: Corrected as requested.
Comment 5: (Page 10, Line 12) The results and discussion does not provide the original spectra of MSI and only adds the table of values.
Response: The requested information is provided as Supplementary Figure.
Comment 6: (Page 10, Line 29) The respective R2 values for the samples packaged under MAP were 0.77, 0.60, and 0.56 is not in conformity with the Table 3.
Response: Corrected.
Comment 7: (Page 12, Table 2) The authors referred to the findings of this work demonstrated that PLS-R models developed on FT-IR data for the evaluation of the microbiological quality of gilthead sea bream fillets packaged under aerobic and MAP conditions presented satisfactory performance in the prediction of the microbial growth in terms of TVC, but the respective R2 value is much lower than the cross-validation R2 value.
Response: The value of R2 in the aerobic storage for cross-validation and prediction is 0.89 and 0.74, respectively. It is expected to have lower R2 value in the prediction set as these data are provided into the model for the first time and there was no previous training of the model with these data. So it is expected to have lower model performance.